# Classifying Dysphagic Swallowing Sounds with Support Vector Machines

**DOI:** 10.3390/healthcare8020103

**Published:** 2020-04-21

**Authors:** Shigeyuki Miyagi, Syo Sugiyama, Keiko Kozawa, Sueyoshi Moritani, Shin-ichi Sakamoto, Osamu Sakai

**Affiliations:** 1Department of Electronic Systems Engineering, Graduate School of Engineering, The University of Shiga Prefecture, Hikone, Shiga 522-8533, Japan; ot23ssugiyama@ec.usp.ac.jp (S.S.); sakamoto.s@e.usp.ac.jp (S.-i.S.); sakai.o@e.usp.ac.jp (O.S.); 2Department of Nutrition, School of Human Cultures, The University of Shiga Prefecture, Hikone, Shiga 522-8533, Japan; kozawa.k@shc.usp.ac.jp; 3Head, Neck, and Thyroid Surgery, Kusatsu General Hospital, 1660, Yabase, Kusatsu, Shiga 525-8585, Japan; suemoritani@gmail.com

**Keywords:** dysphagia, swallowing sound, machine learning, support vector machine (SVM)

## Abstract

Swallowing sounds from cervical auscultation include information related to the swallowing function. Several studies have been conducted on the screening tests of dysphagia. The literature shows a significant difference between the characteristics of swallowing sounds obtained from different subjects (e.g., healthy and dysphagic subjects; young and old adults). These studies demonstrate the usefulness of swallowing sounds during dysphagic screening. However, the degree of classification for dysphagia based on swallowing sounds has not been thoroughly studied. In this study, we investigate the use of machine learning for classifying swallowing sounds into various types, such as normal swallowing or mild, moderate, and severe dysphagia. In particular, swallowing sounds were recorded from patients with dysphagia. Support vector machines (SVMs) were trained using some features extracted from the obtained swallowing sounds. Moreover, the accuracy of the classification of swallowing sounds using the trained SVMs was evaluated via cross-validation techniques. In the two-class scenario, wherein the swallowing sounds were divided into two categories (viz. normal and dysphagic subjects), the maximum F-measure was 78.9%. In the four-class scenario, where the swallowing sounds were divided into four categories (viz. normal subject, and mild, moderate, and severe dysphagic subjects), the F-measure values for the classes were 65.6%, 53.1%, 51.1%, and 37.1%, respectively.

## 1. Introduction

Dysphagia assessment is a clinically paramount task for detecting dysphagia and presbyphagia in patients, as well as preventing or reducing some diseases caused by a swallowing disorder. For example, pulmonary aspiration due to dysphagia is one of the most common causes of pneumonia, which is the third leading cause of death following malignant neoplasms and heart disease in Japan. Previous research has demonstrated that the death rate caused by pneumonia increases with age [1]. Aspiration pneumonia is the primary type of pneumonia that occurs in older people. Based on clinical research, more than 70% of people with pneumonia appear to have aspiration pneumonia [2].

Currently, the video fluoroscopic (VF) swallowing test is the golden standard for assessing dysphagia [3]. However, the equipment for the VF test is expensive, and the medical attendants performing the test may be exposed to radiation. The video endoscopic (VE) swallowing test is a common test in clinics and hospitals [4]. Because the equipment for the VE test is small, VE can be performed at the patient’s bedside. Nevertheless, the disadvantage of both tests is that they require trained clinicians to perform them. In particular, these tests cannot be performed by nurses, speech-language pathologists, or care helpers.

Alternatively, cervical auscultation is one of the most useful methods for dysphagic assessment [5]. The usefulness of cervical auscultation has been demonstrated throughout the literature. For example, Zenner et al. reported that the results obtained by performing a dysphagia examination with cervical auscultation had a high level of agreement with those obtained from the VF test in patients under long-term care [6]. However, to perform cervical auscultation, trained clinicians or speech-language pathologists are required. Nurses and care helpers in clinics or rehabilitation centers cannot perform cervical auscultation. For overcoming such a situation, an easily operable system for cervical auscultation is expected.

To design such a system, one of the key issues is developing a processing method for the signal obtained from cervical auscultation [7].

In the literature, several researchers have proposed different types of signal processing methods for cervical auscultation. Dudik et al. reviewed several past approaches of signal processing for auscultation [8]. Initially, Takahashi et al. demonstrated the possibility of dysphagic screening using a microphone and accelerometer [9]. In addition, automated classification of abnormal swallowing has been investigated in several studies [10,11,12,13,14,15]. The relation between swallowing sounds and the mechanism of swallowing [16], as well as the differences in swallowing sounds among various categories of subjects, such as the young, old, normal, or abnormal subjects [17], have been studied previously. Recently, Dudik et al. demonstrated the performance of a deep brief neural network for distinguishing normal swallowing from that in unhealthy patients [18]. The majority of the proposed methods have been applied to two-class problems. However, a conclusive method for automatically classifying swallowing signals based on multiple dysphagic levels has not been realized.

In this study, the support vector machine (SVM)— a machine learning framework— has been applied to swallowing sounds for classifying the subjects into different dysphagic categories. The classification results were compared with the categories based on the dysphagic assessment performed by clinicians. Subsequently, we evaluated the accuracy of the resulting SVM model in classifying the subjects into different dysphagia categories.

## 2. Method

Healthy subjects, 17 men and 10 women (age range: 21–47, mean: 22.4) with no dysphagic diseases were recruited. Between 2015 and 2017, 78 male patients and 65 female patients (age range: 25–102, mean: 83.3) hospitalized in Kusatsu General Hospital were recruited as dysphagic subjects. The following study protocol received the approval from the ethics committee at the University of Shiga Prefecture and Kusatsu General Hospital.

According to the modified water swallowing test, multiple 3 mL samples of water were given to the subjects. Acoustic sounds were subsequently recorded using a neck-mounted microphone connected to a laptop computer. We used an SH-12iK microphone (Nanzu Musen, Shizuoka, Japan) with a frequency range of 200–3000 Hz. The computer recording software used was Audacity; the sampling rate was set to 8000 Hz, while the recording gain was 0.7. The segment of the swallowing sound was extracted from the successively recorded sound. As the typical swallowing period is known to be about 700 ms in a healthy person [16], a sound segment of a duration of 800 ms was obtained to sufficiently cover most of the swallowing sound. The center of the sound segment corresponded to the position of the peak intensity of the swallowing sound. The total number of segments for the swallowing sound was 170.

Furthermore, clinicians assessed all the healthy subjects and patients according to the VE scoring method proposed by Hyodo et al. [19]. Accordingly, they were categorized into four groups, as listed in Table 1. Category A included healthy subjects. The swallowing sound segments for each subject were also categorized; these results are also listed in Table 1, along with the number of categorized sound segments.

### 2.1. Preprocessing

The preprocessing of the obtained sounds consisted of three steps: noise suppression [20], compensation, and sensitivity of the used microphone. In addition, a filter was applied for the frequency band limit.

In the first step, as the noise profile of the microphone was required for noise suppression, the output sounds obtained from the recording system with no input signals were recorded beforehand in an anechoic chamber. By applying fast Fourier transform (FFT) on the output sounds, the frequency characteristics of the noise was obtained. By subtracting the noise characteristic from the swallowing sound segments in the frequency domain, the silent swallowing sounds were generated.

In the second step, as the sensitivity of the used microphone is required, the amplitude of the recorded signal was measured using the Audacity software. At the beginning, each input had a sinusoidal wave with pressure of 0.1 Pa and a frequency range between 50 Hz–10 kHz in steps of 50 Hz. Fitting a polynomial curve to the variation of the measured amplitude produced the sensitivity level of the microphone, $K_{0}\left[f\right]$. The relative sensitivity, K[f] was derived from the division of $K_{0}\left[f\right]$ using the maximum value of $K_{0}\left[f\right]$. The obtained example of K[f] is demonstrated in Figure 1. By multiplying the inverse of K[f] by the magnitude spectrum of the swallowing sounds, the compensated swallowing sounds were obtained.

Finally, the spectral range of the swallowing sounds was restricted from 200 Hz–3 kHz because of the microphone’s acoustic restriction. Furthermore, applying the inverse FFT to the resulting spectrum yielded the final preprocessed swallowing signals. Figure 2 shows an example of the preprocessed swallowing signal.

### 2.2. Features

In this subsection, we describe the definitions of the various features that were used for the machine learning process. For extracting the features from the frequency and time-frequency domains of the preprocessed sound signal x[n]n=0,1,2,⋯N−1, we applied the discrete time Fourier series and a short time Fourier transform to x[n]. The definitions for both transforms are given as follows: (1)$$\begin{array}{c} X\left[k\right]=\sum_{n=0}^{N-1}x\left[n\right]e^{j\frac{2\pi k}{N}n} \end{array}$$(2)$$\begin{array}{c} X_{w}\left[m,k\right]=\sum_{n=0}^{L-1}x_{m}\left[n\right]e^{j\frac{2\pi k}{L}n}, \end{array}$$
where $x_{m}\left[n\right]$ denotes the windowed signal by applying the window function w[n−mS] with a window size *L*, frame number *m*, and skip width *S* to x[n]. The maximum frame number is defined by M=N/S. A spectrogram defined as $\left|X_{w}\left[m,k\right]\right|$ from (2) is used in the following Section 2.2.2. An example of the spectrogram is shown in Figure 3a.

#### 2.2.1. Frequency Domain Features

The following conventional features associated with X[k] were used.

Maximum magnitude of spectra [13] $A_{\max}\equiv \max_{k=0\cdots N/2}\left|X[k]\right|$.Peak frequency [13] $f_{\max}\equiv \frac{F_{s}}{N}\arg\max_{k=0,\cdots ,N/2}\left|X[k]\right|$, where $F_{s}$ denotes the sampling frequency.Frequency average [17]
$$\bar{f}\equiv \sum_{k=0}^{N/2}\frac{f_{k}\left|X[k]\right|}{\sum_{l}^{N/2}\left|X[l]\right|}$$
where $f_{k}=kF_{s}/N$.Standard deviation of frequency
$$\sigma_{f}\equiv \sqrt{\sum_{k=0}^{N/2}\frac{{\left(f_{k}-\bar{f}\right)}^{2}\left|X[k]\right|}{\sum_{l}^{N/2}\left|X[l]\right|}}.$$Frequency median and quartile ratioBy considering the magnitude of the spectra |X[k]| as part of a frequency distribution, the quartiles $k_{1}$, $k_{2}$, $k_{3}$ can be calculated, where $k_{2}$ corresponds to the median, and the median frequency is given by $Q_{2}=k_{2}F_{s}/N$. Other indices can also be translated into the frequency $Q_{i}=k_{i}F_{s}/N$. Consequently, the quartile ratio is defined as $Q_{r1}\equiv Q_{1}/Q_{2},\;Q_{r3}\equiv Q_{3}/Q_{2}$.Total energy
$$E=\frac{1}{N^{2}}\sum_{k=0}^{N-1}{\left|X[k]\right|}^{2}$$

#### 2.2.2. Time-Frequency Domain Features

In this study, we propose some features derived from the spectrogram $|X_{w}\left[m,k\right]|$ based on which a binary image $X_{\mathrm{th}}\left[m,k\right]$ is obtained via thresholding. These obtained features describe some characteristics of the time-frequency domain signal, including statistical variation, peak locations, and dispersion of peak values.

First, some features associated with the spectrogram in the time-frequency domain are restricted by the indices shown in Figure 4, which are defined as follows.

Maximum magnitude of the spectrogram $A_{w}\equiv \max_{m,\;k}\left|X_{w}\left[m,k\right]\right|$.Peak location of the spectrogram $\left(m_{w},\;k_{w}\right)\equiv \arg\max_{m,\;k}\left|X_{w}\left[m,k\right]\right|$.Relative distances of the peak location from the center of the spectrogram
$$\begin{array}{c} D\equiv \sqrt{D_{H}^{2}+D_{V}^{2}}\\ D_{H}\equiv m_{w}-M/2,\;D_{V}\equiv k_{w}-L/4. \end{array}$$Total energy of the spectrogram $E_{s}\equiv \sum_{m}\sum_{k}{\left|X_{w}\left[m,k\right]\right|}^{2}$.

Furthermore, to describe the time variation of the statistics with respect to the spectrogram, the spectrogram is divided into *B*-blocks along the *m* axis, as displayed in Figure 4; subsequently, the following features are defined in each block. The quartile $k_{b1},\;k_{b2},\;k_{b3}$ for each block is derived from the histogram $H_{b}\left[k\right]$ of the *b*-th block given by
$$H_{b}\left[k\right]\equiv \sum_{m^{\prime }=0}^{M/B-1}\left|X_{w}\left[\left(M/B\right)b+m^{\prime },k\right]\right|.$$

The frequencies for the corresponding quartiles were calculated as $Q_{bi}=k_{bi}F_{s}/L$. The quartile ratio can be similarly defined by $Q_{rb1}\equiv Q_{b1}/Q_{b2},\;Q_{rb3}\equiv Q_{b3}/Q_{b2}$. Each block energy is derived from the square sum of the spectrogram, including each block, as
$$E_{b}=\sum_{m^{\prime }=0}^{M/B-1}\sum_{k=0}^{L/2}{\left|X_{w}\left[\left(M/B\right)b+m^{\prime },k\right]\right|}^{2}.$$

In the later experiments, *B* = 15 is used. This means that each block length is approximately 0.05 s under the current sampling rate. The width of the envelope of each peak in the preprocessed signal shown in Figure 2 can be affordably dropped into the block length. Hence, each block holds sufficient information of the difference signal among the other blocks.

Finally, to describe the effects of multiple spectral peaks and the spread of each peak, we propose the following features with respect to the threshold spectrogram $X_{\mathrm{th}}\left[m,\;k\right]$ defined as
$$X_{\mathrm{th}}\left[m,\;k\right]\equiv \left\{\begin{array}{cc} 1 & X_{w}\left[m,\;k\right]\geq \varepsilon_{t}\\ 0 & \mathrm{otherwise} \end{array}\right.,$$
where $\varepsilon_{t}\equiv \kappa A_{w}$ is a threshold level related to the above peak value *A_w_* in the spectrogram and the scale *κ*. In the following experiments, we used *κ* = 0.5. From Figure 3a, there are some local peaks in the spectrogram, except a central main peak. Applying a higher threshold level to the spectrogram may remove the positions of the other spectral peaks. In the case of a lower threshold level, many peak locations, including noisy level spectral peaks, appear. Preliminary experiments determine that *κ* = 0.5 could give the appropriate peak location shown in Figure 3b.

Area of the threshold spectrogram:
$$A_{\mathrm{th}}\equiv \sum_{k=0}^{M-1}\sum_{k=0}^{L/2}X_{\mathrm{th}}\left[m,\;k\right]$$Ratio of the area of the threshold spectrogram to the complete area of the spectrogram:
$$R_{\mathrm{th}}\equiv A_{\mathrm{th}}/\left(ML/2\right)$$Average distance from the center of the spectrogram:
$$\begin{array}{c} {\bar{D}}_{h}=\frac{1}{A_{\mathrm{th}}}\sum_{k=0}^{M-1}\sum_{k=0}^{L/2}X_{\mathrm{th}}\left[m,\;k\right]\left(m-\frac{M}{2}\right)\\ {\bar{D}}_{v}=\frac{1}{A_{\mathrm{th}}}\sum_{k=0}^{M-1}\sum_{k=0}^{L/2}X_{\mathrm{th}}\left[m,\;k\right]\left(k-\frac{L}{4}\right)\\ \bar{D}=\frac{1}{A_{\mathrm{th}}}\sum_{k=0}^{M-1}\sum_{k=0}^{L/2}X_{\mathrm{th}}\left[m,\;k\right]\sqrt{{\left(m-\frac{M}{2}\right)}^{2}+{\left(k-\frac{L}{4}\right)}^{2}} \end{array}$$Average distance from the peak location of the spectrogram:
$$\begin{array}{c} {\bar{D}}_{ph}=\frac{1}{A_{\mathrm{th}}}\sum_{k=0}^{M-1}\sum_{k=0}^{L/2}X_{\mathrm{th}}\left[m,\;k\right]\left(m-m_{s}\right)\\ {\bar{D}}_{pv}=\frac{1}{A_{\mathrm{th}}}\sum_{k=0}^{M-1}\sum_{k=0}^{L/2}X_{\mathrm{th}}\left[m,\;k\right]\left(k-k_{s}\right)\\ {\bar{D}}_{p}=\frac{1}{A_{\mathrm{th}}}\sum_{k=0}^{M-1}\sum_{k=0}^{L/2}X_{\mathrm{th}}\left[m,\;k\right]\sqrt{{\left(m-m_{w}\right)}^{2}+{\left(k-k_{w}\right)}^{2}} \end{array}$$

### 2.3. Machine Learning

In this subsection, we describe the data set conditions and size adjustments for training the SVMs. Generally, the choice of machine learning method depends on the size of the data set. Deep learning requires vast training data for constructing proper models. For example, the CIFAR-10 data set consists of 60,000 images in 10 classes, with 6000 images per class [21]. It is difficult to collect such a large number of swallowing sounds of patients in a clinical setting. It is experimentally well-known that the SVMs have the advantage of operating with relatively small data sets [22,23]. Therefore, the SVMs were employed in this study. We used the LIBSVM library [24], which is commonly used for implementing SVMs. The efficiency of an SVM with a radial basis function (RBF) kernel depends on the cost and RBF kernel parameter; therefore, these parameters are optimized using the grid-search tool included in the LIBSVM library. The number of swallowing sounds belonging to each category are not always equal to each other. The number of samples from each category used for training should be adjusted to obtain consistent results from the SVM, such that they are almost equal to each other. For example, in the two-class problem, the swallowing sound segments belonging to category A are classified as normal swallowing, and those belonging to categories B, C, and D were classified as abnormal swallowing. Table 1 lists a higher number of sound segments for abnormal swallowing than those for normal swallowing. The number of sound segments that were used was restricted to 104 randomly selected segments. Overall, segments from both groups were divided into 84 training data samples and 20 test data samples. Five different sets of the training and testing data samples were obtained, and all the classified results were averaged to obtain the final classification accuracy. In the four-class problem, the smallest number of the available sound segments was 37 for category D; hence, 37 sound segments were randomly selected from categories A, B, and C. These 37 sound segments were divided into 27 training data samples and 10 test data samples. The remaining process was similar to the two-class problem.

### 2.4. Choice of Features

When parameter *B* is set to 15, a total of 83 features can be obtained. These features can create various combinations of features, which require large computational costs for machine learning. For reducing the number of feature combinations for the computational cost, the following restriction, which is a type of filtering method [25], was introduced. The correlation coefficients between the VE scoring and each feature value were calculated beforehand, and they were sorted into the descending order of the absolute value of the correlation coefficients. This variation of the sorted correlation coefficients is shown in Figure 5. Figure 5 shows that the magnitude of the correlation coefficients rapidly decreases, and the decrease is almost linear after the 10th feature. Therefore, the first nine features were used as the candidate features for the machine learning experiments. The classification accuracy for using the top *k* features of all the 83 features are plotted in Figure 6. The graph is saturated at approximately four features in the two-class problem, and six features in the four-class problem. Therefore, the feature combinations _9_*C*_4_, _9_*C*_6_ were studied in these experiments.

## 3. Results

Table 2 lists the classification accuracy in the two-class problem, where the swallowing sounds of group A are classified as normal, whereas those of groups B, C, and D are classified as abnormal. In this Table, the list has been sorted in descending order of accuracy. The highest value of the accuracy is 0.780, and the highest F-measure is 0.789.

Table 3 summarizes the result of the classification accuracy for the four-class problem. The list is sorted in descending order of total accuracy. The highest total accuracy is 0.460 with features ${\bar{D}}_{v}$, $\bar{f}$, *A_w_*, *Q*_*r*2_, ${\bar{D}}_{h}$, and *Q*_*r*1_. The F-measures for all feature combinations in each class are plotted in Figure 7. The x-axis and y-axis in those figures indicate the combination number and the F-measure value, respectively.

## 4. Discussion

The accuracy in the two-class problem shows similar performance compared to the previous results reported by Mérey et al. [26]. They applied the SVM with an RBF kernel to the dual accelerometry data and demonstrated that the accuracy is 0.806 by using simple feature combination. Based on the feature combination listed in Table 2, the average distance from the center of the spectrogram in the frequency direction, ${\bar{D}}_{v}$, as well as in the quartile-related features, such as *Q*_*r*1_ and *Q*_8,2_, are commonly used for obtaining higher accuracy, which suggests that combining these features is useful for increasing the precision of the classification performance. All of these features depend on the position of the frequency peaks or the spectrum bias. As Lee pointed out in [13], the presence of vallecular residue and pyriform sinus residue may change the pharyngeal air space volume, which may cause a change in the acoustic vibration characteristics of the airflow in the pharynx. The feature value of the above features reflects this change.

In the four-class problem, the feature ${\bar{D}}_{v}$ and quartile-related features are also included in the combination of the features from Table 3. Although combination #42 yields the highest accuracy and holds a relatively high F-measure for classes A, B, and C, its F-measure in class D is less than 0.15 from Figure 7. Combination #70 indicates the highest F-measure of 0.371 within class D (see Figure 7d). Similarly, combinations #61 and #75 hold the second and third highest F-measures of 0.362 and 0.356, respectively. These feature combinations commonly include ${\bar{D}}_{h}$ rather than ${\bar{D}}_{v}$, which is included in the feature combinations with high accuracy. These results suggest the following:The characteristic of swallowing sounds for class D is different from that of the other classes.The feature ${\bar{D}}_{h}$ relates to the area position with a higher magnitude along the time direction in the spectrograms. Robbins et al. reported that the total duration of oropharyngeal swallowing was significantly longer in the oldest group than in any other younger counterparts [27]. Borr et al., also reported that the duration of the onset time, and deglutition apnea of auscultation for the dysphagic group, was significantly longer than that of the nondysphagic group [28]. These duration variances caused a temporal bias in the spectrogram.When classifying the swallowing sounds into four categories, multiple classifiers or a combination of these might be useful due to the different characteristics of each class.

## 5. Conclusions

The accuracy of classifying swallowing sounds into two classes and four classes by using SVMs was examined. Although the F-measure reached 78.9% in the two-class scenario, the overall classification accuracy of the four-class scenario was still insufficient when using the classifier as a stand-alone method for the diagnosis. Conversely, this study revealed that the variation of some feature combinations can serve as useful classifiers for individual categories.

Although these results demonstrate the potential of using a classifier constructed by the SVM, some problems and limitations remain in their use as a practical classifier. The results of the trained SVM relied on the category defined by the total score obtained by Hyodo’s VE scoring. The scoring method consisted of four tests. The segmented swallowing sounds may not correspond to the results of the total score of the four tests. To solve this problem, an improved model for considering each test in the scoring should be created. For selecting the features for the SVMs, a filter-based method was used in the experiments and the restriction on the feature combination was introduced. The resulting feature combination may not be optimal. Other types of selection methods, such as wrapper methods or embedded methods, should also be applied and the resulting classifier performances should be compared with each other. It will also be useful to employ other types of machine learning for increasing the classification performance. Deep neural networks is one such option, provided many data sets can be obtained. We are currently working on constructing an improved classifier for considering each test based on Hyodo’s VE scoring by using deep neural networks combined with data augmentation. We will report the results of this research in the future.

## Figures and Tables

**Figure 1 healthcare-08-00103-f001:**
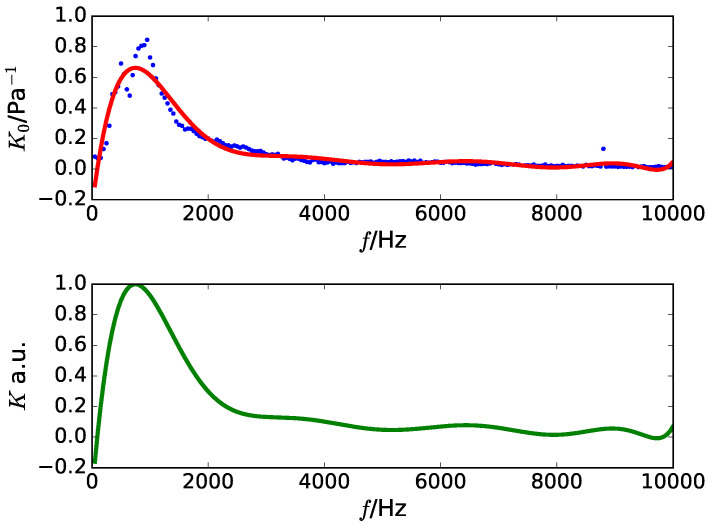
Example of the sensitivity calculation of a microphone.

**Figure 2 healthcare-08-00103-f002:**
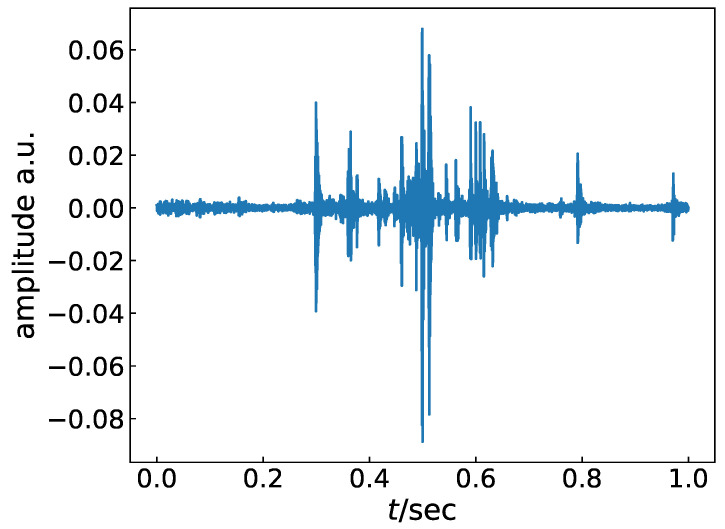
Example of a preprocessed swallowing signal.

**Figure 3 healthcare-08-00103-f003:**
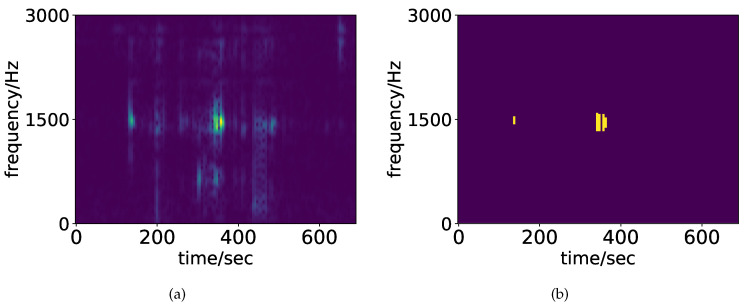
Example of a spectrogram (**a**) and its threshold version (**b**).

**Figure 4 healthcare-08-00103-f004:**
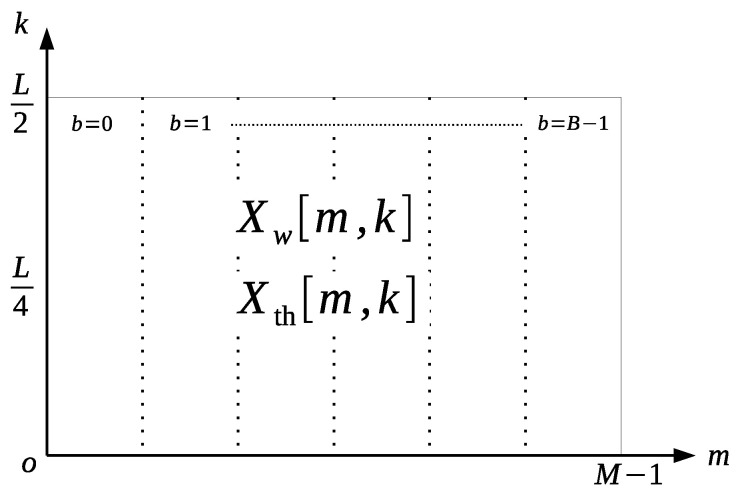
Definition of blocks in the time-space domain.

**Figure 5 healthcare-08-00103-f005:**
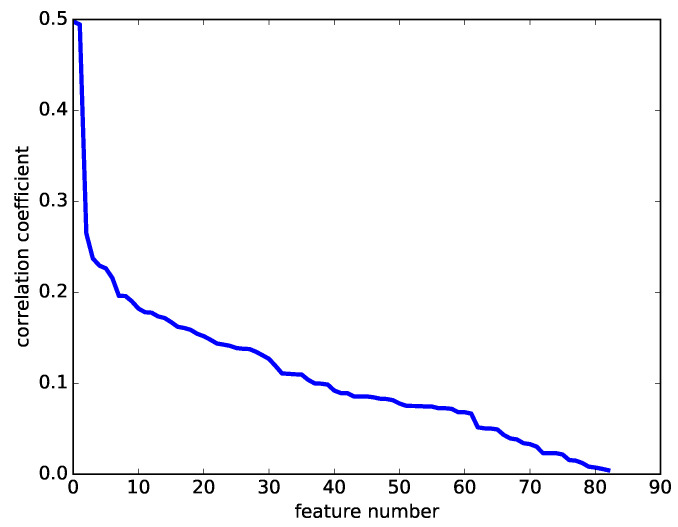
The variation of the correlation coefficients for the total VE scoring performed by the clinicians.

**Figure 6 healthcare-08-00103-f006:**
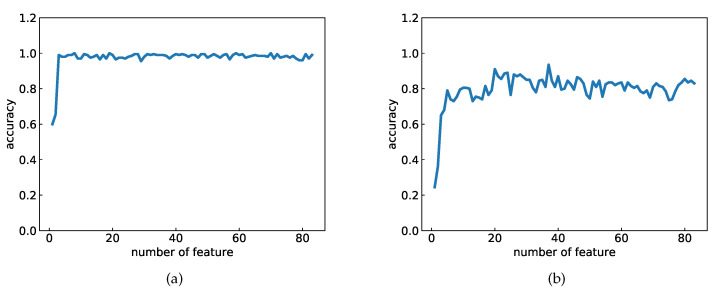
Variation in classification accuracy by using top *k* features in (**a**) the two-class problem and (**b**) four-class problem.

**Figure 7 healthcare-08-00103-f007:**
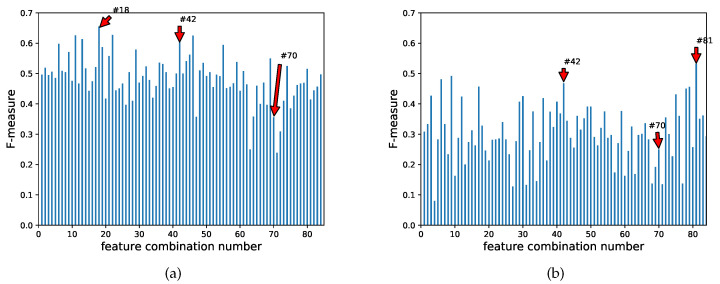
F-measure for each feature combination in classes A, B, C, and D. (**a**) Class A, (**b**) Class B, (**c**) Class C, (**d**) Class D.

**Table 1 healthcare-08-00103-t001:** Definition of dysphagia classification categories derived from scoring based on the method proposed by Hyodo et al. [19] using the VE swallowing test.

Category	Total Score Range	Dysphagic Level	Number of Sounds
A	0	normal	104
B	1–4	mild	66
C	5–9	moderate	214
D	10–12	severe	37

**Table 2 healthcare-08-00103-t002:** Accuracy, precision, recall, and F-measure for each feature combination in the two-class problem. The list is sorted in descending order of accuracy.

Comb. #	Used Features				Accuracy	Precision	Recall	F-Measure
8	${\bar{D}}_{v}$	$\bar{D}$	*A* _*w*_	*Q* _2_	0.780	0.787	0.790	0.781
18	${\bar{D}}_{v}$	$\bar{D}$	*Q* _2_	*Q* _8,2_	0.780	0.804	0.750	0.771
54	${\bar{D}}_{v}$	*Q* _2_	${\bar{D}}_{h}$	*Q* _8,2_	0.780	0.831	0.710	0.760
48	${\bar{D}}_{v}$	*A* _max_	*Q* _2_	*Q* _*r*1_	0.775	0.811	0.720	0.762
113	*A* _*w*_	*A* _max_	*Q* _2_	*Q* _*r*1_	0.770	0.763	0.790	0.773
34	${\bar{D}}_{v}$	$\bar{f}$	${\bar{D}}_{h}$	*Q* _*r*1_	0.770	0.752	0.820	0.781
46	${\bar{D}}_{v}$	*A* _*w*_	*Q* _*r*1_	*Q* _8,2_	0.770	0.737	0.870	0.789
53	${\bar{D}}_{v}$	*Q* _2_	${\bar{D}}_{h}$	*Q* _*r*1_	0.770	0.750	0.810	0.779
19	${\bar{D}}_{v}$	$\bar{D}$	${\bar{D}}_{h}$	*Q* _*r*1_	0.755	0.717	0.870	0.780
17	${\bar{D}}_{v}$	$\bar{D}$	*Q* _*r*2_	*Q* _*r*1_	0.750	0.808	0.670	0.722

**Table 3 healthcare-08-00103-t003:** Accuracy and F-measures of classes A, B, C, and D for each feature combination in the four-class problem. The list has been sorted in descending order of accuracy.

Comb. #	Used Features					
42	${\bar{D}}_{v}$	$\bar{f}$	*A_w_*	*Q* _2_	${\bar{D}}_{h}$	*Q* _*r*1_
3	${\bar{D}}_{v}$	$\bar{D}$	$\bar{f}$	*A_w_*	*A* _max_	*Q* _*r*1_
29	${\bar{D}}_{v}$	$\bar{D}$	*A_w_*	*Q* _*r*2_	*Q* _*r*1_	*Q* _8,2_
17	${\bar{D}}_{v}$	$\bar{D}$	$\bar{f}$	*Q* _2_	${\bar{D}}_{h}$	*Q* _*r*1_
6	${\bar{D}}_{v}$	$\bar{D}$	$\bar{f}$	*A_w_*	*Q* _2_	*Q* _*r*1_
46	${\bar{D}}_{v}$	$\bar{f}$	*A* _max_	*Q* _2_	${\bar{D}}_{h}$	*Q* _*r*1_
9	${\bar{D}}_{v}$	$\bar{D}$	$\bar{f}$	*A_w_*	${\bar{D}}_{h}$	*Q* _8,2_
45	${\bar{D}}_{v}$	$\bar{f}$	*A_w_*	${\bar{D}}_{h}$	*Q* _*r*1_	*Q* _8,2_
11	${\bar{D}}_{v}$	$\bar{D}$	$\bar{f}$	*A* _max_	*Q* _2_	${\bar{D}}_{h}$
75	$\bar{D}$	*A_w_*	*A* _max_	${\bar{D}}_{h}$	*Q* _*r*1_	*Q* _8,2_
**Comb. #**		**Accuracy**	**F-Measure (A)**	**F-Measure (B)**	**F-Measure(C)**	**F-Measure(D)**
42		0.460	0.602	0.468	0.511	0.145
3		0.420	0.495	0.427	0.500	0.133
29		0.420	0.579	0.407	0.313	0.338
17		0.415	0.521	0.457	0.459	0.095
6		0.410	0.598	0.481	0.79	0.174
46		0.410	0.625	0.360	0.333	0.208
9		0.405	0.571	0.492	0.259	0.141
45		0.400	0.562	0.256	0.400	0.256
11		0.390	0.626	0.288	0.396	0.143
75		0.385	0.385	0.431	0.355	0.356

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
