# Peer review of "Classifying Dysphagic Swallowing Sounds with Support Vector Machines"

_healthcare, 2020, doi:10.3390/healthcare8020103_

Round 1
Reviewer 1 Report
- Classification of swallowing sounds and vibrations is a well-researched problem, and there are a number of significant contributions in this field. The authors have not properly researched the literature in order to understand the state-of-the-art contributions.
Reviewer 2 Report
In this paper, authors evaluated the adaptability of SVM for classifying the degree of dysphagia with swallowing sounds and proposed a greedy feature selection method based on the correlation coefficients between VE score proposed by Hyodo et al. and each feature which authors utilized. In addition, authors showed the high classification performance of the proposed method in the empirical experiments. I think the proposed method is simple yet efficient and effective for classifying dysphagic swallowing, though I am not familiar with healthcare area.
However, for publication, I have the following concerns and unclear points. Please clarify and explain them in more detail.
1) In this research, authors did not mention the age range of dysphagic subjects. Swallowing performance is generally reduced depending on age. Therefore, the classification of swallowing for elder subjects is a more difficult task. The information of subjects is very important. Please show the age range of patients.
2) In addition to condition 1), authors employed the healthy subjects whose age is 22-23 in this research. In general, to reduce the effect of age factor, subjects of various age range should be employed. Why do authors employ only such young healthy subjects? Please explain the intent of employment in more detail.
3) Why do authors select SVM? Many classification methods in machine learning have existed. Please explain the motivation for selection in more detail.
4) In this research, authors proposed a feature selection method. Although a combination of all features causes the classification performance reduction due to noisy features, the combination (without feature selection) has the largest information. Therefore, to demonstrate the feature selection performance, authors should show the classification accuracy with the combination of all features as a baseline. 83 is not so large number in machine learning area. Please show the classification accuracy of it.
5) In this paper, authors change from the problem that the optimal combination of features from 83 features to the problem that the combination from 9 features which are selected according to the correlation between VE scoring and each feature. Similar to the condition 3), many optimization techniques are existed in machine learning such as GA. Why did authors directly investigate the optimal feature selection from 83 features? The appropriateness of the selected 9 features for classifying the dysphagic swallowing is unclear. Please explain the motivation that authors select the greedy approach instead of other optimization techniques.
6) In this paper, some magic numbers are existed such as B = 15, \kappa = 0.5, etc. How to decide these numbers? Please explain in more detail.
7) In the experiment of the four-class problem, authors evaluated 9C6=84 feature combinations. However, in Figure 5, I counted the vertical bars, I could not find only 82 or 83 vertical bars (#83 may be overlapped with the vertical axis of graph?). Where are #83 and #84? Please check it and replace the correct figures.
Reviewer 3 Report
I couldn't tell the age of the patient's with dysphagia. This isn't clearly identified in lines 57-61. Overall, I enjoyed the paper but wanted to see a more thorough results and conclusion section. I felt that the authors didn't identify the limitations of this study, implications, and future directions of research.
Round 2
Reviewer 2 Report
Authors clarified the motivation of utilizing SVM and greedy feature selection approach, and the information of subjects. In addition, authors showed the experimental results with all features and the all combination of selected features.
I think my previous concerns are clarified.
Reviewer 3 Report
Thank you for addressing my previously expressed concerns. I believe this topic is important to our field. Keep up the good work.